# Changes in Sedentary and Active Lifestyle, Diet Quality and Body Composition Nine Months after an Education Program in Polish Students Aged 11–12 Years: Report from the ABC of Healthy Eating Study

**DOI:** 10.3390/nu11020331

**Published:** 2019-02-03

**Authors:** Lidia Wadolowska, Jadwiga Hamulka, Joanna Kowalkowska, Natalia Ulewicz, Monika Hoffmann, Magdalena Gornicka, Monika Bronkowska, Teresa Leszczynska, Pawel Glibowski, Renata Korzeniowska-Ginter

**Affiliations:** 1Department of Human Nutrition, University of Warmia and Mazury in Olsztyn, Sloneczna 45F, 10-718 Olsztyn, Poland; lidia.wadolowska@uwm.edu.pl (L.W.); joanna.kowalkowska@uwm.edu.pl (J.K.); 2Department of Human Nutrition, Faculty of Human Nutrition and Consumer Sciences, Warsaw University of Life Science—SGGW, Nowoursynowska 159 C, 02-776 Warsaw, Poland; jadwiga_hamulka@sggw.pl (J.H.); magdalena_gornicka@sggw.pl (M.G.); 3Department of Functional Food, Ecological Food and Commodities, Faculty of Human Nutrition and Consumer Sciences, Warsaw University of Life Science—SGGW, Nowoursynowska 159C, 02-776 Warsaw, Poland; monika_hoffmann@sggw.pl; 4Department of Human Nutrition, Faculty of Food Science, Wroclaw University of Environmental and Life Science, Pl. Grunwaldzki 24A, 50-363 Wroclaw, Poland; monika.bronkowska@upwr.edu.pl; 5Department of Human Nutrition, Faculty of Food Technology, University of Agriculture in Krakow, Balicka 122, 30-149 Krakow, Poland; teresa.leszczynska@urk.edu.pl; 6Department of Biotechnology, Microbiology and Human Nutrition, Faculty of Food Science and Biotechnology, University of Life Sciences in Lublin, Skromna 8, 20-704 Lublin, Poland; pawel.glibowski@up.lublin.pl; 7Department of Commodity Science and Quality Management, Gdynia Maritime University, Morska 81-87, 81-225 Gdynia, Poland; r.ginter@wpit.am.gdynia.pl

**Keywords:** adolescents, adiposity, central obesity, dietary patterns, diet quality, overweight, physical activity, sedentary time, screen time, pre-teenagers

## Abstract

The sustainability of education focused on improving the dietary and lifestyle behaviours of teenagers has not been extensively studied. The aim of this study was to determine the sustainability of diet-related and lifestyle-related school-based education on sedentary and active lifestyle, diet quality and body composition of Polish pre-teenagers in a medium-term follow-up study. An education-based intervention study was carried out on 464 students aged 11–12 years (educated/control group: 319/145). Anthropometric measurements were taken and body mass index (BMI) and waist-to-height ratios (WHtR) were calculated, both at the baseline and after nine months. Dietary data from a short-form food frequency questionnaire (SF-FFQ4PolishChildren) were collected. Two measures of lifestyle (screen time, physical activity) and two diet quality scores (pro-healthy, pHDI, and non-healthy, nHDI) were established. After nine months, in the educated group (vs. control) a significantly higher increase was found in nutrition knowledge score (mean difference of the change: 1.8 points) with a significantly higher decrease in physical activity (mean difference of the change: −0.20 points), nHDI (−2.3% points), the z-WHtR (−0.18 SD), and the z-waist circumference (−0.13 SD). Logistic regression modelling with an adjustment for confounders revealed that after nine months in the educated group (referent: control), the chance of adherence to a nutrition knowledge score of at least the median was over 2 times higher, and that of the nHDI category of at least the median was significantly lower (by 35%). In conclusion, diet-related and lifestyle-related school-based education from an almost one-year perspective can reduce central adiposity in pre-teenagers, despite a decrease in physical activity and the tendency to increase screen time. Central adiposity reduction can be attributed to the improvement of nutrition knowledge in pre-teenagers subjected to the provided education and to stopping the increase in unhealthy dietary habits.

## 1. Introduction

Currently, the most popular sedentary behaviour of teenagers is time spent in front of TV, computer and tablet screens, etc. (screen time) [1,2,3]. In children and adolescents, it was shown that a higher screen time increases the chance of unhealthy food choices, poor dietary habits and obesity [2,4,5,6]. The potential mechanism may be twofold. High parents’ screen time is known to influence children’s screen time [7]. Secondly, children who have higher screen time are less physically active [8,9,10]. Regular walking (e.g., to school) is a recommended type of activity for schoolchildren, with a daily step count of 10,000 [11]. The latest Polish study reports that 21.5% of children aged 9–17 years reach the World Health Organization (WHO) recommended levels of physical activity (PA) for health, with significantly more boys (28.5%) being active than girls (15.2%) [12]. In 2013, the Polish Ministry of Sport and Tourism established the campaign “Stop skipping physical education classes” to promote PA among schoolchildren and encourage them to attend physical education (PE) classes at school. The target groups include children and adolescents themselves, along with parents, PE teachers, schools, local governments and doctors [13].

Since dietary choices develop early, a school-based nutrition education program may be an effective way to prevent childhood obesity by lowering modifiable obesity risk factors [14]. Interestingly, the changes in dietary habits and PA were more pronounced among teenagers with reduced body weight than with those normal body weight [15]. In Polish school-aged children, common poor dietary habits include breakfast skipping, high consumption of sweets and low consumption of fruit and vegetables [16,17]. Poor dietary habits are more often reported in girls, children of parents with lower education level, those living in the countryside and those with worse household economic situation and housing conditions. Unhealthy dietary habits in adolescence can predict metabolic syndrome in adulthood, especially higher waist circumference (WC), which was shown to be the best anthropometric measure of adiposity [18].

Given all of this, education programs have been focused on diet or physical activity, mainly considered separately, with outcomes assessed in a short-term follow-up [19]. To date, most education programs have focused on reducing body weight in overweight and obese adolescents [20,21,22]. Verloigne et al. [23] showed that an important role is attributed to parents, suggesting that parents should be involved in education programs. However, Baghurst et al. [14] showed that nutrition education within school programs positively impacts dietary choices, irrespective of parental input. Furthermore, programs designed to increase youth self-efficacy could play a role in preventing childhood obesity [14]. A review of 14 dietary programs found that only 6 of them revealed significant changes in adiposity indices or could be considered successful in affecting childhood obesity [6]. In addition, within these six interventions, two were focused on dietary behaviours and four were focused on both dietary behaviours and PA [6]. The outcomes were measured in various terms after intervention, but longer-term sustainability of interventions is weakly known. Nevertheless, even if a lack of changes in dietary habits does not cause weight gain or central obesity, education programs are recommended for a higher diet quality [24,25].

The sustainability of education focused on improving teenagers’ dietary and active/inactive lifestyle behaviours from a longer-term perspective has not yet been studied extensively. The aim of this study was to determine the sustainability of diet-related and lifestyle-related school-based education on sedentary and active lifestyle, diet quality and body composition for Polish pre-teenagers in a medium-term follow-up study.

## 2. Materials and Methods

### 2.1. Study Design

The study was designed as an education-based intervention study with a convenience sampling without random allocation into experimental (educated) and control groups. School classes were assigned to either the intervention or the control group, and all students from these classes were invited. The school was the first selection unit. In selecting schools, we tried to ensure that schools assigned to the educated and the control group were similarly located (rural, suburban, urban) and with a similar number of students.

The study was performed by academic researchers from seven Polish universities in eight locations covering the entire territory of Poland. The research had two sources of funding: for the educated group, the “ABC of Healthy Eating” project; for the control group, the Polish Ministry of Sciences and Higher Education (see Funding section). The data were collected by researchers in June 2015 (at baseline) and in March 2016 (at a nine-month follow-up). The researchers were well trained in taking anthropometric measures and collecting dietary data. More details on the study protocol and methods were described previously [26].

In the educated group, a diet-related and lifestyle-related school-based education program lasting three weeks was implemented. The education program covered five topics lasting a total of 15 h and was provided as talks and workshops focused on activating participants. Each topic lasted 4 h of school lessons (approximately 180 min) and included various forms of education from fun to “scientific” cognition (Appendix A). The education program was provided in schools by a minimum of 3 or 4 academic researchers involved in the study. School teachers were not involved in the education program, but they were present during educational activities. Apart from the study, all students took part in the regular school activities containing some content related to nutrition and a healthy lifestyle. More details on the education program were described previously [26]. The effect of education was measured nine months (±2 weeks) after the baseline.

### 2.2. Participants

Recruitment was conducted in selected elementary schools from urban, suburban and rural areas. Details of sample selection were described previously [26]. In brief, students (with an expected age of 11–12 years) from fourth- and fifth-grade classes were invited to attend. It was decided to start recruitment based on school classes because students were subject to the same school education and would be at a similar stage of development. School inclusion criteria were (i) a location at a convenient distance from academic centres (up to 50 km) and (ii) the agreement of the school principal for the school to participate. The school exclusion criterion was previous participation of the school in other nutrition–health education programs. Participant inclusion criteria were (i) the written consent of parents or legal guardians to participate, (ii) being in a fourth- or fifth-grade class of elementary school and (iii) age 11 or 12 years. The participant exclusion criterion was disability self-declared by a parent, legal guardian or teacher.

In total, 48 classes were selected across Poland (Figure 1). Initially, 668 students were recruited. Of these, 208 participants were excluded from analyses—17 participants because of age below 11 years or above 12 years and 191 participants because of absence from some stages of the study. In total, the study included 464 adolescents (educated/control group: 319/145), at baseline aged 11–12 years, comprising 216 boys (46.6%) and 248 girls (53.4%).

Details regarding the participants’ socio-demographic characteristics, nutrition knowledge, sedentary and active lifestyle, diet quality and body composition at the baseline and at a nine-month follow-up are given in Table 1.

Sample size was calculated in regard to nutritional knowledge as the main objective of the project, based on data collected before. Details were described previously [26]. In brief, based on the expected increase in nutrition knowledge score at the nine-month follow-up and the expected difference between the groups (educated vs. control), the sample size required was 304 (200 vs. 104) respondents. For this calculation, a 5% significance level and 80% power were considered. Next, for the current study, the adequacy of the study sample (educated/control: 319/145) was checked by applying the post hoc statistics to calculate power. For the data under study, the statistical power was, for example, 3%–51% for overweight/obesity and 43%–93% for central obesity. Thus, based on the calculations and taking into account the precision of the methods used, we found that the sample size was sufficient to detect differences between groups, if they exist.

### 2.3. Data Collection

A Short Form of the Food Frequency Questionnaire for Polish Children (SF-FFQ4PolishChildren) was used to collect data related to diet, sedentary and active lifestyle, nutrition knowledge and sociodemographic characteristics. Details regarding the questionnaire have been described previously [26]. The questionnaire was self-administered by pre-teenagers in the classroom and supervised by researchers. Explanations were given if necessary. The completion of the questionnaire took the pre-teenagers approximately 40 min.

### 2.4. Nutrition Knowledge

Nutrition knowledge went beyond the interests of the current study but was monitored at baseline and at a nine-month follow-up. The nutrition knowledge score was determined on the basis of 18 questions. Participants were asked about nutrition based on questions developed by Whati et al. [29] and adapted to Polish conditions and education (Appendix A). Correct answers were scored with 1 point, while wrong or “I don’t know” or missing answers were scored with 0 points. For each participant, points were summed up to calculate their nutrition knowledge score (range: 0 to 18 points). Based on variable distribution (at baseline; a posteriori approach), two levels of nutrition knowledge score were established above (inclusive) and below the median to classify respondents as those with higher nutrition knowledge (≥6.0 points) and those with lower nutrition knowledge (<6.0 points).

### 2.5. Sedentary and Active Lifestyle

Two measures of sedentary and active lifestyle were considered: screen time and physical activity.

Screen time was assessed using the question “How much time do you spend watching TV or on the computer or in front of a computer on an average day of the week?” The participants could choose one of six answers (with assigned scores): <2 h/day (0 points), ≥2 and <4 h/day (1 point), ≥4 and <6 h/day (2 points), ≥6 and <8 h/day (3 points), ≥8 and <10 h/day (4 points), or ≥10 h/day (5 points). For each participant, screen time expressed in points was calculated. After combining some answers, based on variable distribution (at baseline; mixed a priori and a posteriori approach), two categories of screen time were established to classify respondents as those with lower screen time (<4 h/day) and those with higher screen time (≥4 h/day).

PA was assessed using two questions regarding PA at school and during leisure time. The participants could choose one of three answers describing their PA at school (low, moderate, vigorous) and during leisure time (low, moderate, vigorous). Many examples for each answer were given (Table 2). Finally, after combining some categories of both questions (a priori approach), the participants were divided into three PA levels: low, moderate and high, with assigned scores from 0 points to 5 points. For each participant, PA expressed in points was calculated. Vigorous PA at school (most of the time related to high physical exertion) combined with vigorous PA during leisure time (activities requiring physical effort for over 3 h/week) was categorised as high PA and was considered as adhering to WHO recommendations on PA [30], although the criterion used was not exactly the same as that given by WHO experts. The WHO recommends that children and teenagers aged 5–17 years have a minimum of 2.5 h/day of moderate-intensity PA. Finally, two categories of PA were considered: those with adherence to the WHO recommendation for PA and those without adherence. 

### 2.6. Diet Quality

Diet quality scores were established based on the usual food frequency consumption of eight food items within the 12 last months. For food frequency consumption, participants could choose one of seven categories (converted into daily frequency, times/day; Table 3).

To create both diet quality scores, food items were selected based on previous knowledge and similar studies (a priori approach) [31,32]. Two opposite diet quality scores were used to fully describe diet, especially for those participants who were engaged in a variety of opposite behaviours:a pro-Healthy Diet Index (pHDI) which included four food items: dairy products, fish, vegetables and fruit;a non-Healthy Diet Index (nHDI) which included four food items: fast foods, sweetened carbonated drinks, energy drinks and sweets or confectionery.

Each diet quality score was calculated by summing up the daily frequencies of food items (range: 0 to 8 times/day) and recalculated to percentages (range: 0% to 100%) because a range of 0–100 units is easier to interpret. Higher percentage reflected higher adherence to the diet quality score, i.e., for pHDI, a better diet quality, and for nHDI, a worse diet quality. Based on variable distribution (at baseline; a posteriori approach), two levels of each diet quality score were established above (inclusive) and below the median to classify respondents:pHDI: ≥25.875% (more pro-healthy diet) and <25.875% (less pro-healthy diet);nHDI: ≥11.625% (more unhealthy diet) and <11.625% (less unhealthy diet).

### 2.7. Body Composition

The measurements of body weight (kg), height (cm) and waist circumference (WC, cm) were taken, all recoded with a precision of 0.1 kg or 0.1 cm, respectively, using professional devices and measuring tape. All measurements were taken in light clothing and without shoes according to the guidelines [33]. Body mass index (BMI, kg/m^2^) and waist-to-height ratio (WHtR) were calculated.

Body composition characteristics were interpreted using direct and indirect approaches. BMI and WHtR were categorised by applying international standards (direct approach) [27,28]. BMI-for-age was categorised according to sex-specific BMI cut-offs for teenagers given by the International Obesity Task Force (IOTF) in 2012 [28] as follows:thinness: BMI-for-age < 18.5 kg/m^2^;normal weight: BMI-for-age 18.5 to 24.9 kg/m^2^;overweight/obesity (as a general adiposity measure): BMI-for-age ≥ 25 kg/m^2^.

WHtR of ≥0.5 was used as a central obesity (central adiposity) measure according to Ashwell et al. [27].

Z-scores of BMI, WHtR and WC were calculated to achieve a mean equal to 0 and standard deviation (SD) equal to 1. The position of each individual within a group at baseline (indirect approach) was determined and changes of the position after nine months were monitored. Based on variable distribution (at baseline; a posteriori approach), z-scores were categorised as follows: <−1 SD, −1 to 1 SD and >1 SD (a posteriori approach) to classify subjects into lower, normal and higher levels of each measure, respectively [34,35].

### 2.8. Confounders

Gender, age, residence and family affluence were considered as confounders.

The Family Affluence Scale (FAS) was derived from the household characteristics. The study used a scale composed of four questions described by the Polish team of the Health Behaviour of School-aged Children (HBSC) study [36]. For each answer, points (from 0 to 2) were assigned. The points were summed up for each participant (range 0 to 7 points). Details have been described previously [26].

### 2.9. Statistical Analysis

Categorical variables were presented as a sample percentage (%) and continuous variables as a mean with a 95% confidence interval (95% CI). The differences between groups (educated vs. control or baseline vs. follow-up) were verified with a chi-square test (categorical variables) or a Mann–Whitney test (continuous variables). Before statistical analysis, the normality of variable distribution was checked with a Kolmogorov–Smirnov test.

Logistic regression modelling was used to assess the chance of falling in the nutrition knowledge, sedentary and active lifestyle, diet quality and body composition categories (i) at a nine-month follow-up with respect to the baseline as a reference and (ii) in the educated group with respect to the control group as a reference. The odds ratios (ORs) and 95% CIs were calculated. A crude model and a model with an adjustment for gender, age (years), residence (rural, urban) and FAS (points) were created. For body composition characteristics, additional confounders were included in the adjusted model: pHDI (%), nHDI (%), screen time (points) and physical activity (points). The significance of ORs was assessed by Wald’s statistics. For all tests, *p* < 0.05 was considered significant. Analyses were performed using Statistica software (version 12.0 PL; StatSoft Inc., Tulsa, OK, USA; StatSoft, Krakow, Poland).

## 3. Results

### 3.1. Nutrition Knowledge Changes

After nine months in the educated group (vs. control), a higher increase in nutrition knowledge score was found (mean difference of the change: 1.8 points, *p* < 0.0001) (Table 4). In the educated group (vs. control), the adherence to the nutrition knowledge score category above (inclusive) the median was significantly higher by 93% (*p* < 0.01; adjusted model) at baseline and over 2 times higher (*p* < 0.001; adjusted model) after nine months (Table 5).

### 3.2. Sedentary and Active Lifestyle Changes

After nine months, in the educated group (vs. control), a higher decrease in PA was found (mean difference of the change: −0.20 points, *p* < 0.05) (Table 4). No significant difference in change in screen time between groups (educated vs. control) after nine months was found (mean difference of the change: −0.01 points, insignificant (NS)). When both groups were considered separately, there were no significant changes in PA after nine months in the educated (mean change by −0.21 points, NS) or control group (mean change by −0.01 points, NS). Furthermore, there were no significant changes in screen time after nine months in the educated (mean change by 0.12 points, NS) or control group (mean change by 0.13 points, NS).

In the educated group (vs. control), the chance of adherence to the WHO recommendation on PA was significantly higher by 74% (*p* < 0.05; adjusted model) at baseline, but insignificant after nine months (Table 5). In the educated group (vs. control), the chance of screen time ≥4 h/day was insignificant at baseline and significantly lower by 41% (*p* < 0.05; adjusted model) after nine months.

### 3.3. Diet Quality Changes

After nine months in the educated group (vs. control), a significant decrease in the nHDI was revealed (mean difference of the change: −2.3 percentage points, *p* < 0.05) (Table 4). When both groups were considered separately, there were no significant changes in the nHDI in the educated group (mean change −0.2 percentage points, NS) and the control group (mean change 2.1 percentage points, NS). No significant difference in the change in pHDI between the educated and the control group after nine months was found.

After nine months in the educated group (vs. control), the chance of adherence to the nHDI category above (inclusive) the median was significantly lower by 35% (*p* < 0.05; adjusted model) (Table 5). After nine months in the educated group (vs. control), the chance of adherence to the pHDI category above (inclusive) the median did not differ significantly.

### 3.4. Body Composition Changes

After nine months in the educated group (vs. control), a higher decrease in the z-WHtR (mean difference of the change: −0.18 SD, *p* < 0.001) and the z-waist circumference (mean difference of the change: −0.13 SD, *p* < 0.05) was found (Table 4). When both groups were considered separately, after nine months there were no significant changes (follow-up vs. baseline) in the z-WHtR and the z-waist circumference in either the educated group or the control group. No significant differences in the z-BMI-for-age between groups (educated vs. control) were found after nine months.

In the educated group (vs. control), at baseline, the chance of central obesity was over 5 times higher (adjusted model; *p* < 0.01) and the chance of z-waist circumference > 1 SD was over two times higher (adjusted model; *p* < 0.05), but after nine months both ORs lowered (from 5.24 to 2.02 and 2.20 to 1.55, respectively; adjusted models) and the significance of these associations disappeared (Table 5). In the educated group (vs. control), the chance of overweight/obesity at baseline or after nine months was significantly lower by 40% or 39% (both *p* < 0.05; adjusted models), respectively. In the educated group (vs. control), the chance of a z-waist circumference of <−1 SD at baseline or after nine months was significantly lower by 79 % (*p* < 0.0001; adjusted model) or 68 % (*p* < 0.001; adjusted model), respectively.

The results of logistic regression modelling for nutrition knowledge, sedentary and active lifestyle, diet quality scores and body composition considered separately in the educated and the control group are presented in Appendix A, while mean changes in the characteristics under study after nine months (follow-up vs. baseline) in the total sample are presented in Appendix A.

## 4. Discussion

It was found that after nine months, the risk of central obesity was lowered for students who were included in the education program in comparison to students who were out of the program. In the educated group, a greater reduction in PA than in the control group was revealed. In all students, included in the education program or not, the tendency to a similar increase in screen time was observed. After nine months, there was a higher increase in nutrition knowledge and a decrease in unhealthy dietary habits in the educated group, while no difference was found for pro-healthy dietary habits between the groups.

The reduction of the central obesity occurrence nine months following implementation of the education program shows that such programs can be effective in obesity prevention. It is worth underlining that in the educated group, reduction in two central adiposity measures (WHtR and WC) was shown. On the other hand, no change in general adiposity measure (BMI) was noted. It can be assumed that pre-teenagers from the educated group experienced favourable changes in body composition, i.e., an increase in lean body mass. This supposition requires further research with the use of more accurate methods of body composition assessment. Furthermore, there is strong evidence that WHtR is a better measure of adiposity and all-cause mortality than BMI [27]. Thus, based on previous findings and our results, a clear conclusion in regards to a reduction in the central obesity risk can be drawn. These results support those of other studies showing that education programs activating participants and aimed at improving both diet and lifestyle, rather than those only focused on diet, are more successful in obesity prevention [14,37]. Moreover, our results show the benefit of such a school program targeted at younger pre-teenagers in general, irrespective of their body weight status and without parental input. Therefore, taking into account pre-teenagers’ self-efficacy and increase in feeling of independence with age (from 8 to 12 years old), future school-based diet-related and lifestyle-related education programs can be designed for pre-teenagers to effectively support optimal health [14,38,39].

Despite the implementation of an education program related to active/inactive lifestyle, after nine months we found a greater decline in PA in the educated than the control group, and in all participants, there was a similar tendency toward an increase in sedentary behaviour. This proves the lack of sustainability of the program in increasing pre-teenagers’ physical activity despite an increase in the nutrition knowledge of pre-teenagers who have been included in the education program [19,40]. On the other hand, a positive effect in the prevention of central obesity was achieved. This result is not easy to explain. It should be assumed that the reduction in central obesity can be attributed to factors other than active/inactive lifestyle. It is thought that nowadays, a general reduction in PA is unavoidable with teenagers’ increasing age [9,10,41]. The main reasons include the increase in schoolwork and school duties, especially in Polish vs. European elementary schools, the pressure of peers and/or the fashion of using electronic devices which is accompanied by an increase in time spent in front of the screen and a gain in sedentary behaviours. A decline in PA with age has been reported in teenagers across Europe, including in Poland [42,43]. Our results are in line with those found by Sorenson et al. [44]. The authors found no significant changes in PA (in minutes/day) four months after implementation of an education program. Mitchell et al. [5] found that sedentary time increased from 12 to 16 years old. In a pooled analysis consisting of 26 longitudinal studies, it was found that PA decreases by 7% per year during adolescence [41]. Recent evidence by Fakhouri et al. [8] suggests that screen time and PA are not associated in children aged 6–11 years. The current study indirectly showed that in pre-teenagers aged 11–12 years there may exist a negative link between sedentary and active behaviour, i.e., the more time spent before the screen, the lower the PA level. In contrast, Biddle et al. [45] reported that the association between sedentary behaviour and adiposity in youth is not clear and claims of causality are premature.

The study suggests that a reduction in central obesity can be attributed to dietary factors. Surprisingly, no increase in pro-healthy dietary habits was found in pre-teenagers who were included in the education program in comparison to those outside of the education program. Thus, in this area, the education program failed. In contrast, in teenagers (14–19 years), a significant improvement in the acquisition of healthy eating habits nine months after nutritional education was shown [46]. A possible explanation for our findings is well-established family dietary habits that are difficult to change without the involvement of adults as well as food availability at home or school, which, for pre-teenagers, largely depends on adults. Taken together, this suggests that to improve the pro-healthy dietary habits of pre-teenagers, activities should be directed at the adults who are responsible for children’s nutrition.

An interesting finding of the study is that there was no increase in unhealthy dietary habits in the educated group and, to the contrary, a tendency to increase in the control group. The first possible explanation is that an education program could stop the increase in unhealthy dietary habits with age. Such unfavourable changes in dietary behaviours with age have been widely reported in European and American teenagers [47,48,49]. Among Czech adolescents aged 11–15 years, non-participation in organised leisure-time activities was an indicator of unhealthy dietary habits and lifestyle [50]. In our study, unhealthy dietary habits were related to fast food consumption frequency, sweetened carbonated drinks, energy drinks and sweets or confectionery. It is thought that avoiding these foods could be easier and such decisions were more dependent on teenagers than adults. In contrast, the increase in pro-healthy dietary habits (in our study, those related to dairy products, fish, vegetables and fruit) required the involvement of adults to provide children with access to these foods. This strengthens the previous statement that improving the pro-healthy dietary habits of young teenagers depends on the involvement of adults [37,51]. The second possible explanation is that the education program was sustainable after nine months with respect to unhealthy dietary habits and not sustainable with respect to pro-healthy dietary habits [40,52]. It may be supposed that this positive change in pro-healthy dietary habits can be sustainable only in the short term, and this effect quickly disappears if the education is not repeated.

### Strengths and Limitations

The sample was relatively large (over 460), taking into account that in the nine-month follow-up, complete dietary and weight status data were obtained at all stages of the study. We applied a very rigorous selection of the sample; each participant had to take part in all meetings and workshops covering five topics of the education program lasting 3 weeks—students absent at least one school day at the meeting or workshop were excluded. The next strength of the study is the use of appropriate statistical methods to interpret the differences between the educated and control groups with regard to the change from baseline to the nine-month follow-up. Finally, we used simple measures of general and central adiposity, appropriate for a comprehensive assessment of high adiposity levels [29], although in further studies the use of more accurate measures of body composition is recommended.

The main limitation is the lack of random subject allocation into educated and control groups. We were unable to apply the random approach for several reasons. First, for organisational reasons, we wanted to choose schools located a convenient distance from the academic centres (up to 50 km). Secondly, (surprisingly) many school principals did not permit their school to participate in the study. Thirdly, many of the schools had previously participated in other nutrition–health education programs, so they could not be included in our study. There was an element of randomness in our study because assigning classes to the educated or control group was random and the schools represented a wide social cross section covering the entire territory of Poland. Finally, due to the differences between the baseline educated and control groups, several methods of statistical analysis were used. In the study, physical activity was not directly measured. We used self-reported data which could have caused errors in estimating the changes in physical activity after nine months and in detecting differences between the educated group and the control group. On the other hand, in epidemiological studies, many questionnaires (international and country-specific) are widely used to measure physical activity level in adolescents and adults, e.g., International Physical Activity Questionnaire (IPAQ) and Dietary Habits and Nutrition Beliefs Questionnaire (KomPAN) developed for Poles [31,53,54]. In the current study, we used a questionnaire tested in school-aged children (data unpublished—paper in preparation). The good reproducibility of questions related to screen time and physical activity at school or during leisure time was found (kappa statistics: 0.46–0.54; total agreement (the same category in test and retest): 64.8–74.6% in 11–15-year-old respondents). Thus, we think that the error resulting from the use of self-reported data to estimate changes in sedentary and active lifestyle is relatively small.

## 5. Conclusions

Diet-related and lifestyle-related school-based education, assessed over almost one year, can reduce central adiposity in pre-teenagers, despite the decrease in physical activity and the tendency to increase time spent in front of the screen. The reduction in central adiposity can be attributed to improvement of nutrition knowledge in pre-teenagers provided education and to stopping the increase in unhealthy dietary habits. Since an increase in pro-healthy dietary habits was not achieved, in this area, the education program failed.

To prevent obesity, in dietary–lifestyle education addressed to pre-teenagers, special emphasis should be placed on the elimination of unhealthy dietary habits. However, the shaping of pro-healthy dietary habits and improving lifestyle by an increase in physical activity and a decrease in screen time should not be overlooked. We suppose that to strengthen the pro-healthy dietary habits of pre-teenagers, education programs should be implemented addressed to the adults who are responsible for children’s nutrition.

## Figures and Tables

**Figure 1 nutrients-11-00331-f001:**
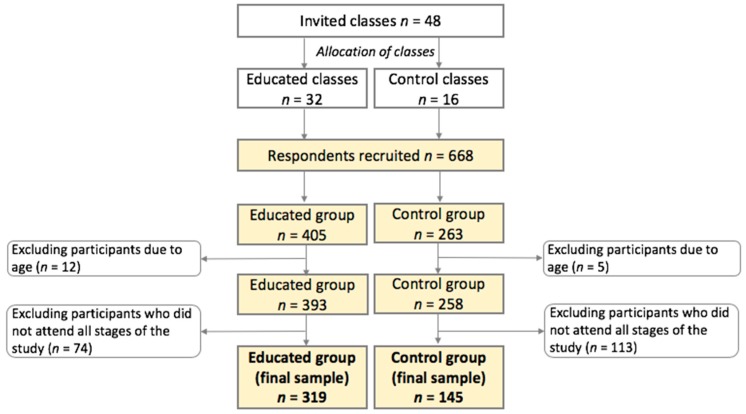
Sample collection. Notes: cream colour indicates respondents and/or study groups.

**Table 1 nutrients-11-00331-t001:** Sample baseline characteristics and at a nine-month follow-up.

	Baseline				Follow-Up			
	Total Sample	Educated Group	Control Group	*p*-Value	Total Sample	Educated Group	Control Group	*p*-Value
Sample size, *n* (%)	464 (100.0)	319 (100.0)	145 (100.0)		464 (100.0)	319 (100.0)	145 (100.0)	
Gender, *n* (%)				ns				
Boys	216 (46.6)	141 (44.2)	75 (51.7)					
Girls	248 (53.4)	178 (55.8)	70 (48.3)					
Age (years), mean (95% CI)	11.9 (11.9, 12.0)	11.9 (11.9, 11.9)	12.0 (11.9, 12.0)	ns	12.9 (12.9, 13.0)	12.9 (12.9, 12.9)	13.0 (12.9, 13.0)	ns
Place of residence, *n* (%)				ns				
Rural	162 (34.9)	102 (32.0)	60 (41.4)					
Urban	302 (65.1)	217 (68.0)	85 (58.6)					
Family Affluence Scale (points), mean (95% CI)	5.3 (5.1, 5.5)	5.3 (5.1, 5.5)	5.3 (5.1, 5.6)	ns	5.5 (5.3, 5.6)	5.5 (5.3, 5.7)	5.5 (5.2, 5.7)	ns
Nutrition knowledge score (points), mean (95% CI)	6.0 (5.7, 6.2)	6.1 (5.9, 6.4)	5.5 (5.1, 6.0)	*	7.9 (7.7, 8.2)	8.3 (7.9, 8.6)	7.2 (6.8, 7.6)	***
Nutrition knowledge score ≥ Me, *n* (%)	253 (54.5)	189 (59.2)	64 (44.1)	**	372 (80.5)	268 (84.0)	104 (72.7)	**
Screen time (points), mean (95% CI)	0.84 (0.74, 0.94)	0.81 (0.69, 0.92)	0.92 (0.74, 1.09)	ns	0.97 (0.87, 1.06)	0.93 (0.81, 1.05)	1.05 (0.88, 1.22)	ns
Screen time ≥ 4 h/day, *n* (%)	87 (18.8)	53 (16.7)	34 (23.4)	ns	109 (23.5)	66 (20.8)	43 (29.7)	*
Physical activity (points), mean (95% CI)	3.65 (3.53, 3.78)	3.76 (3.62, 3.90)	3.42 (3.19, 3.66)	*	3.5 (3.4, 3.6)	3.5 (3.4, 3.7)	3.4 (3.2, 3.7)	ns
Adherence to WHO recommendation on physical activity, *n* (%)	153 (33.0)	114 (35.7)	39 (27.1)	ns	125 (27.1)	82 (25.9)	43 (29.7)	**
pHDI (% points), mean (95% CI)	27.7 (26.4, 29.0)	28.5 (27.0, 30.0)	25.8 (23.3, 28.3)	*	28.1 (26.8, 29.5)	28.9 (27.3, 30.5)	26.4 (24.1, 28.7)	ns
pHDI ≥ Me, *n* (%)	235 (50.9)	167 (52.5)	68 (47.2)	ns	239 (51.8)	171 (53.9)	68 (47.2)	ns
nHDI (% points), mean (95% CI)	14.3 (13.3, 15.3)	14.2 (13.0, 15.4)	14.6 (13.0, 16.3)	ns	14.8 (13.8, 15.9)	14.0 (12.8, 15.2)	16.7 (14.7, 18.7)	*
nHDI ≥ Me, *n* (%)	233 (50.2)	153 (48.0)	80 (55.2)	ns	239 (51.7)	154 (48.4)	85 (59.0)	*
Central obesity, *n* (%)	45 (10.0)	40 (13.0)	5 (3.5)	**	46 (9.9)	37 (11.6)	9 (6.2)	ns
BMI-for-age, *n* (%)				*				
thinness	46 (10.2)	32 (10.4)	14 (9.8)		43 (9.3)	33 (10.4)	10 (6.9)	*
normal weight	286 (63.4)	206 (66.9)	80 (55.9)		287 (62.1)	205 (64.7)	82 (56.6)	
overweight/obesity	119 (26.4)	70 (22.7)	49 (34.3)		132 (28.6)	79 (24.9)	53 (36.6)	
z-Waist circumference, *n* (%)				****				
<−1 SD	57 (12.6)	21 (6.8)	36 (25.2)		64 (13.8)	30 (9.4)	34 (23.4)	***
−1 to 1 SD	333 (73.8)	237 (76.9)	96 (67.1)		328 (70.8)	233 (73.3)	95 (65.5)	
>1 SD	61 (13.5)	50 (16.2)	11 (7.7)		71 (15.3)	55 (17.3)	16 (11.0)	

Sample size may vary in variables due to missing data. Family Affluence Scale (range: 0–7 points); Nutrition knowledge score (range: 0–18 points); Me: median; Median for Nutrition knowledge score: 6.0 points; Screen time (range: 0–5 points); Physical activity (range: 0–5 points); Adherence to the WHO recommendations on physical activity (PA) was considered as meeting both vigorous PA at school (most of the time related to high physical exertion) and vigorous PA at leisure time (activities requiring physical effort over 3 h/week)—details are given in Table 1; pHDI: pro-Healthy Diet Index (range: 0%–100%); Median for pHDI: 25.875%; nHDI: non-Healthy Diet Index (range: 0%–100%); Median for nHDI: 11.625%; Central obesity identified as waist-to-height ratio of ≥0.5 according to Ashwell et al. [27]; BMI-for-age categorised with sex-specific cut-offs according to the International Obesity Task Force (IOTF) standards [28], as follows: thinness BMI < 18.5 kg/m^2^; normal weight BMI = 18.5–24.9 kg/m^2^; overweight/obesity BMI ≥ 25 kg/m^2^; Statistically significant (Mann–Whitney test for means or chi-square test for percentage distribution): * *p* < 0.05, ** *p* <0.01, *** *p* < 0.001, **** *p* < 0.0001; ns: not significant.

**Table 2 nutrients-11-00331-t002:** Categorising and scoring (points) of physical activity based on physical activity at school and during leisure time.

Physical Activity at School	Physical Activity during Leisure Time		
	Low	Moderate	Vigorous
Low	Low (0)	Low (1)	Moderate (2)
Moderate	Low (1)	Moderate (3)	Moderate (4)
Vigorous	Moderate (2)	Moderate (4)	High (5) Adherence to WHO recommendation

Physical activity at school was classified as low (most of the time in a sitting position, in class or on breaks), moderate (half the time in a sitting position and half the time in motion), or vigorous (most of the time on the move or in classes related to high physical exertion). Physical activity during leisure time was classified as low (more time spent sitting, watching TV, in front of a computer, reading, light housework, short walks totalling up to 2 h a week), moderate (walking, cycling, gymnastics, working at home or other light physical activity performed for 2–3 h/week), or vigorous (cycling, running, working at home or other sports activities requiring physical effort for over 3 h/week).

**Table 3 nutrients-11-00331-t003:** Converting categories of food frequency consumption into daily frequencies.

Categories of Food Frequency Consumption	Daily Frequency (Times/Day)
never or almost never	0
rarely once a week	0.06
once a week	0.14
2–4 times/week	0.43
5–6 times/week	0.79
every day	1
several times a day	2

**Table 4 nutrients-11-00331-t004:** Means and mean changes (95% CI) in nutrition knowledge, sedentary and active lifestyle, diet quality and body composition in educated and control groups at a nine-month follow-up.

	Baseline			Follow-Up			Change: Follow-Up—Baseline		
	Educated Group	Control Group	Difference	Educated Group	Control Group	Difference	Educated Group	Control Group	Difference
Nutrition knowledge score (points)	6.1 (5.9, 6.4)	5.5 (5.1, 6.0)	0.6 *	8.5 (8.2, 8.8)	6.2 (5.7, 6.6)	2.3 ****	2.4 (2.0, 2.7)	0.6 (0.3, 0.9)	1.8 ****
Screen time (points)	0.81 (0.69, 0.92)	0.92 (0.74, 1.09)	−0.11	0.93 (0.81, 1.05)	1.05 (0.88, 1.22)	−0.12	0.12 (0.02, 0.23)	0.13 (−0.03, 0.29)	−0.01
Physical activity (points)	3.76 (3.62, 3.90)	3.42 (3.19, 3.66)	0.34 *	3.55 (3.41, 3.68)	3.42 (3.17, 3.67)	0.13	−0.21 (−0.34, −0.07)	−0.01 (−0.16, 0.14)	−0.20 *
pHDI (% points)	28.5 (27.0, 30.0)	25.8 (23.3, 28.3)	2.7 *	28.9 (27.3, 30.5)	26.4 (24.1, 28.7)	2.5	0.5 (−1.2, 2.3)	0.8 (−1.6, 3.2)	−0.3
nHDI (% points)	14.2 (13.0, 15.4)	14.6 (13.0, 16.3)	−0.4	14.0 (12.8, 15.2)	16.7 (14.7, 18.7)	−2.7 *	−0.2 (−1.5, 1.1)	2.1 (0.6, 3.7)	−2.3 *
z-WHtR (SDs)	0.16 (0.05, 0.26)	−0.32 (−0.48, −0.15)	0.48 ***	0.08 (−0.03, 0.19)	−0.20 (−0.37, −0.03)	0.28 *	−0.08 (−0.15, −0.01)	0.10 (0.04, 0.16)	−0.18 ***
z-BMI-for-age (SDs)	−0.09 (−0.19, 0.02)	0.18 (0.01, 0.36)	−0.27 **	−0.10 (−0.21, 0.00)	0.23 (0.05, 0.40)	−0.33 ***	−0.01 (−0.07, 0.04)	0.03 (−0.01, 0.07)	−0.04
z-Waist circumference (SDs)	0.16 (0.05, 0.27)	−0.34 (−0.50, −0.19)	0.50 ****	0.11 (0.01, 0.22)	−0.25 (−0.41, −0.09)	0.36 ***	−0.05 (−0.12, 0.02)	0.08 (0.02, 0.14)	−0.13 *

Sample size may vary in variables due to missing data. Nutrition knowledge score (range: 0–18); Screen time (range: 0–5 points); Physical activity (range: 0–5 points); pHDI: pro-Healthy Diet Index (range: 0%–100%); nHDI: non-Healthy Diet Index (range: 0%–100%). The meaning of the terms: Difference (between groups) is related to educated vs. control; Change (in time) is related to follow-up vs. baseline. Statistically significant (Mann–Whitney test): * *p* < 0.05, ** *p* < 0.01, *** *p* < 0.001, **** *p* < 0.0001.

**Table 5 nutrients-11-00331-t005:** Odds ratios (95% CIs) for nutrition knowledge, sedentary and active lifestyle, diet quality and body composition in the educated group at baseline and at a nine-month follow-up.

	Control Group	Educated Group			
		Crude Model		Adjusted Model	
		At Baseline	Follow-Up	At Baseline	Follow-Up
Nutrition knowledge score ≥ Me (ref.: <Me)	ref.	1.84 ** (1.24, 2.74)	1.97 ** (1.23, 3.17)	1.93 ** (1.28, 2.91)	2.15 ** (1.32, 3.51)
Screen time ≥ 4 h/day (ref.: <4 h/day)	ref.	0.66 (0.40, 1.07)	0.62 * (0.40, 0.97)	0.62 (0.38, 1.02)	0.59 * (0.37, 0.94)
Adherence to WHO recommendation on physical activity (ref.: no adherence)	ref.	1.50 (0.97, 2.31)	0.83 (0.53, 1.28)	1.74 * (1.10; 2.73)	0.91 (0.58, 1.42)
pHDI ≥ Me (ref.: <Me)	ref.	1.24 (0.83, 1.84)	1.31 (0.88, 1.95)	1.09 (0.71, 1.66)	1.16 (0.76, 1.77)
nHDI ≥ Me (ref.: <Me)	ref.	0.75 (0.50, 1.11)	0.65 * (0.44, 0.97)	0.79 (0.53, 1.19)	0.70 (0.46, 1.06)
Central obesity (ref.: lack)	ref.	4.12 ** (1.59, 10.70)	1.99 (0.93; 4.25)	5.24 ** (1.96, 14.05)	2.02 (0.85, 4.81)
Thinness (ref.: normal)	ref.	0.89 (0.45, 1.75)	1.32 (0.62, 2.81)	0.96 (0.45, 2.05)	1.23 (0.53, 2.71)
Overweight/obesity (ref.: normal)	ref.	0.55 ** (0.35, 0.87)	0.60 * (0.39, 0.92)	0.60 * (0.38, 0.96)	0.61 * (0.39, 0.96)
z-Waist circumference < −1 SD (ref.: −1 to 1 SD)	ref.	0.24 **** (0.13, 0.43)	0.36 **** (0.24, 0.55)	0.21 **** (0.11, 0.40)	0.32 *** (0.18, 0.58)
z-Waist circumference > 1 SD (ref.: −1 to 1 SD)	ref.	1.84 (0.92, 3.70)	1.40 (0.76, 2,57)	2.20 * (1.05, 4.63)	1.55 (0.82, 2.97)

Sample size may vary in variables due to missing data. Me: median; Median for Nutrition knowledge score: 6.0; Adherence to the WHO recommendations on physical activity (PA) was considered as meeting both vigorous PA at school (most of the time related to high physical exertion) and vigorous PA during leisure time (activities requiring physical effort over 3 h/week)—details are given in Table 1; pHDI: pro-Healthy Diet Index; Median for pHDI: 25.875%; nHDI: non-Healthy Diet Index; Median for nHDI: 11.625%; Central obesity identified as waist-to-height ratio ≥ 0.5 according to Ashwell et al. [27]; Thinness, overweight/obesity and normal weight identified as BMI-for-age categorised by sex-specific cut-offs according to the International Obesity Task Force (IOTF) standards [28], as follows: thinness BMI < 18.5 kg/m^2^; normal weight BMI = 18.5 to 24.9 kg/m^2^; overweight/obesity BMI ≥ 25 kg/m^2^. Odds ratios were adjusted for confounders (at baseline or follow-up, respectively). Adjustment for nutrition knowledge, screen time, adherence to WHO recommendation on physical activity, pHDI and nHDI was as follows: gender, age (years), residence (categorical variable), Family Affluence Scale (points); adjustment for central obesity, thinness, overweight/obesity, z-waist circumference < −1 SD and z-waist circumference > 1 SD was as follows: confounders as mentioned above + screen time (points), physical activity (points), pro-Healthy Diet Index (%), non-Healthy Diet Index (%). Statistically significant (Wald’s statistics): * *p* < 0.05, ** *p* < 0.01, *** *p* < 0.001, **** *p* < 0.0001.

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
