# Peer review of "Changes in Sedentary and Active Lifestyle, Diet Quality and Body Composition Nine Months after an Education Program in Polish Students Aged 11–12 Years: Report from the ABC of Healthy Eating Study"

_nutrients, 2019, doi:10.3390/nu11020331_

Round 1

Reviewer 1 Report

This paper reviews the change in lifestyle behavior and body weight after a 9 months education program.

of the project within the title.

Title: Suggest changing the title and including the name

Line 54: Change Nowadays to currently.

Line 71: Will this paper focus on rural children?

Line 90-91: Is this a family based education or school based?

Methods: Line 48—Please provide more information in the manuscript to justify that combining vigorous activity in school and leisure indicates that participant is meeting PA guidelines.

Table 1:  Provide the potential ranges for the measures in the legend. The table should stand alone.

Results: The authors state “insignificantly increase” in the results. If the results are not significantly different, then there is no change.

Discussion:  The lack of measured increase in physical activity could be due to the measure used to measure physical activity.  The authors could have missed changes in physical activity due to the self-reported method of measuring.

Author Response

*Corresponding author: 

Natalia Ulewicz, MSc

University of Warmia and Mazury

Department of Human Nutrition

Sloneczna 45f, 10-718 Olsztyn, Poland

Tel. +48 (89) 524 5514

E-mail: natalia.ulewicz@uwm.edu.pl

Authors’ Response to the Reviewers’ Comments

Journal:                      Nutrients 

Manuscript:                nutrients-400591

Title:                           Changes in sedentary and activelifestyle, diet quality and body compositionnine months after education program in Polish students aged 11-12 years: report from the ABC of Healthy Eating study.

Authors:                       Lidia Wadolowska, Jadwiga Hamulka, Joanna Kowalkowska, Natalia Ulewicz, Monika Hoffmann, Magdalena GornickaMonika Bronkowska, Teresa Leszczynska, Pawel Glibowski and Renata Korzeniowska-Ginter

Article type:                 research article

17thDecember 2018

Dear Ms. Maryann He, Dear Reviewer,

We are very grateful that we have the opportunity to resubmit the revised version of our manuscript (ID: nutrients-400591) entitled ‘Changes in sedentary and activelifestyle, diet quality and body composition nine months after education program in Polish students aged 11-12 years: report from the ABC of Healthy Eating study’.

We greatly appreciate the time and efforts taken by the Reviewer and the Editor to review our manuscript. We have addressed all issues indicated in the review report, and believe that the revised version can meet the journal publication requirements. 

Please find our responses to the Reviewer’s comments attached. The manuscript has been corrected for language errors, using professional editing (native speaker) and proof-reading service. All changes in the manuscript are highlighted in blue font. 

Yours Sincerely,

Natalia Ulewicz

Reviewer 2 Report

Wadolowska et al. aimed to determine the effects of diet-related and lifestyle-related education on lifestyle, diet quality and body weight. The main conclusions of the study are that education of Polish teenagers can improve dietary choices, leading to decreased consumption of unhealthy foods. However, lifestyle education did not result in an increase in physical activity. Also the subjects did not increase intake of healthy foods, which the authors speculate was due to lack of parental inclusion in the study design.

Although the results are somewhat interesting, it is difficult to see how they will advance the field. Also the authors used a variety of approaches to analyze the data. While they set out to measure body weight, the results presented emphasize differences in the z-scores for waist circumference, a measure that is much less direct.

Overall the tables are difficult to read and lack a uniform mode of presentation. They would benefit from a clearer explanation of which values were compared where significant differences were detected.

The conclusions follow logically from the authors' interpretation of the data.

A minor concern is that there a numerous grammar and spelling errors. Example include the use of was instead of were in line 71 and the use of obesity, a noun, instead of obese, the adjective in line 79.

Author Response

(The authors gave the same response as above.)

Reviewer 3 Report

This paper is reporting on the Polish ABC of Healthy Eating Project and is entitled “Changes in lifestyle, diet quality and body weight nine months after education program in Polish students aged 11-12 years.”

It’s a valuable contribution to the research field dealing with lifestyle behaviours of young people and the role of the school. However the paper as presented suffer from a number of weaknesses that require some work.

My first main concern is that the text is written with quite a number of statistical assumptions that is not properly explained for non statisticians. The majority of readers I would assume come from the health promotion area and this audience would certainly benefit from more explanations and detailing of what was done – and why.

My other main concern is that this paper rests on the assumption of the power of education. Conceptually this is founded on the idea of KAP or KAB models - Knowledge, Attitude, Practice/Behaviour. These assumption are contested, yet there are no explanations/discussions about this issue.  Furthermore reading around line 110 very little detail is given about the nature of the intervention. Keeping the limitations of the KAP/KAB approach in mind I would expect the didactic/curricular nature of the education intervention to be of outmost importance. For instance is it based on a deductive or inductive approach? Experiential and/or problem based?  It might very well be that the nature of the authors approach addresses some of the short comings of the KAB/KAP mindset but much more detail needs to be given  

Following comments will hopefully be instrumental for improving the paper that as a large scale school based life style intervention deserves attention.

Line 41 The text reads “After nine months, in the educated´ group a significantly higher decrease compared to the control group was found in physical activity (mean difference of the change: -0.20 points), the z-waist circumference (-0.13 SD), the z-WHtR (-0.18 SD), nHDI (-2.3 %points).” I am confused: why was a decrease seen and why was it higher in the intervention group when the opposite could be expected”

In line 77 it is stated that “Given all of this, education programs have been mainly focused on diet and healthier dietary habits and less on PA”. I don’t  agree. Is this based on a review of literature or is it the impression of authors? I am convinced that a search for PA education interventions would reveal a huge amount og studies

In line 89 it is stated that “The lasting effect of education focused on improving teenagers dietary and lifestyle behaviours has not yet been studied extensively”. I would recommend to refer to the term “sustainability of intervention” – se for instance the work of O´Loughlin et al. Is the point here that – like in most intervention programs – effects seems to evaporate after the research team has gone back to the lab? In that case the text should be developed to explain that.

Line 96. Design of the Intervention? The text says  that “Analyses were carried out using data from the ABC of Healthy Eatingnational multicenter project (1st edition: ABC of Healthy Nutrition) and the research of centres involved in the study carried out in parallel with the project”. What is the difference between the project and the research? Do you intend to make a distinction between the intervention and the research: Could make sense but needs to be clearly explained.

Line 96 The text says that “Analyses were carried out using data from the ABC of Healthy Eating program. It gives the impression that the paper deals with data that was already analyzed and published. If that’s the case what was then the results and what new does the current paper adds to the field?

Line 100. The text refer to ”well-trained researchers”. In what respect? What kind of training?

Line 102 The text reads that “Sample size was calculated in regards to nutritional knowledge as the main objective of the.” I assume that the author did a kind of quantitative nutritional knowledge score was made and that the distribution of that score is known from a previous study?

Line 110 The text says “The study was designed as an education-based intervention study with a convenience sampling without random allocation”. Was the schools picked totally “a la convenience” or was there any attempts to match the intervention with the non intervention schools?

I wonder about the use of “lifestyle”. Do the authors  exclude diet from that?

Line 95 Typo in ”Participants and study deisgn”

115 The authors mention that “Recruitment was conducted in elementary schools from urban, sub-urban and rural area” What was the assumptions on possible differences between the three? I assume that I was not considered a Confounding factor? (from line 177)

Line 102 – 109 The whole section is difficult to follow and relies on examples. Suggest to rewrite and expand

Line 125. The figure text should be used to explain the figure. The inclusion and exclusion criteria should be explained in the body of the text and not in the boxes.

Line 125 The figure refers to group and classes. The same?

Line 125 It is not clear how the program handled cases where not all students was eligible. In other words was the intervention clustered in the sense that one class was one cluster or what?

Line 154 The details on the pro-Healthy Diet Index (pHDI)and a non-Healthy Diet Index is explained. What was the relationship between the two. Directly inversely related I assume? In that case why is two different measures needed?

From around lines 147, 161 and 171 I get the impression that some kind of categorization takes place according to diet, PA, BMI etc? In other words that students are put into “good”, “medium” etc boxes. If I am right this needs to be explained including the underlying assumptions

Line 193 The text reads “between groups”. What groups. Please explain

Line 208 The text says “Details regarding the participants gender, age, residence, family affluence…………………” I would not see this as a result but simply as a descriptive picture of the sample: As such I suggest it for methods instead

Author Response

*Corresponding author: 

Natalia Ulewicz, MSc

University of Warmia and Mazury

Department of Human Nutrition

Sloneczna 45f, 10-718 Olsztyn, Poland

Tel. +48 (89) 524 5514

E-mail: natalia.ulewicz@uwm.edu.pl

Authors’ Response to the Reviewers’ Comments

Journal:                      Nutrients 

Manuscript:                nutrients-400591

Title:                           Changes in sedentary and activelifestyle, diet quality and body compositionnine months after education program in Polish students aged 11-12 years: report from the ABC of Healthy Eating study.

Authors:                       Lidia Wadolowska, Jadwiga Hamulka, Joanna Kowalkowska, Natalia Ulewicz, Monika Hoffmann, Magdalena GornickaMonika Bronkowska, Teresa Leszczynska, Pawel Glibowski and Renata Korzeniowska-Ginter

Article type:                 research article

17thDecember 2018

Dear Ms. Maryann He, Dear Reviewer,

We are very grateful that we have the opportunity to resubmit the revised version of our manuscript (ID: nutrients-400591) entitled ‘Changes in sedentary and activelifestyle, diet quality and body composition nine months after education program in Polish students aged 11-12 years: report from the ABC of Healthy Eating study’.

We greatly appreciate the time and efforts taken by the Reviewer and the Editor to review our manuscript. We have addressed all issues indicated in the review report, and believe that the revised version can meet the journal publication requirements. 

In relation to suggestions from Reviewer, we would like to uderline that issues regaring study sample and methods were previously published by Hamulka et al. Nutrients, 2018, 10, 1439, doi:10.3390/nu10101439, although we followed all Reviewers suggestions and added more details.

Please find our responses to the Reviewer’s comments attached. The manuscript has been corrected for language errors, using professional editing (native speaker) and proof-reading service. All changes in the manuscript are highlighted in blue font. 

Yours Sincerely,

Natalia Ulewicz

Round 2

Reviewer 2 Report

This manuscript has been greatly approved. I can now support its application.

All the best.

Reviewer 3 Report

A good job done and my recommendation would be to accept it without further changes.